# Association of Combined Sero-Positivity to *Helicobacter pylori* and *Streptococcus gallolyticus* with Risk of Colorectal Cancer

**DOI:** 10.3390/microorganisms8111698

**Published:** 2020-10-30

**Authors:** Meira Epplein, Loïc Le Marchand, Timothy L. Cover, Mingyang Song, William J. Blot, Richard M. Peek, Lauren R. Teras, Kala Visvanathan, Yu Chen, Howard D. Sesso, Anne Zeleniuch-Jacquotte, Sonja I. Berndt, John D. Potter, Marc D. Ryser, Christopher A. Haiman, Sylvia Wassertheil-Smoller, Lesley F. Tinker, Tim Waterboer, Julia Butt

**Affiliations:** 1Cancer Control and Population Health Sciences Program, Duke Cancer Institute and Department of Population Health Sciences, Duke University, Durham, NC 27705, USA; marc.ryser@duke.edu; 2Epidemiology Program, University of Hawai’i Cancer Center, Honolulu, HI 96813, USA; loic@cc.hawaii.edu; 3Department of Medicine and Department of Pathology, Microbiology and Immunology, Vanderbilt University Medical Center, Nashville, TN 37232, USA; timothy.l.cover@vumc.org (T.L.C.); richard.peek@vumc.org (R.M.P.); 4Veterans Affairs Tennessee Valley Healthcare System, Nashville, TN 37212, USA; 5Department of Epidemiology, Harvard T.H. Chan School of Public Health, Boston, MA 02215, USA; mis911@mail.harvard.edu (M.S.); hsesso@hsph.harvard.edu (H.D.S.); 6Department of Nutrition, Harvard T.H. Chan School of Public Health, Boston, MA 02215, USA; 7Clinical and Translational Epidemiology Unit and Division of Gastroenterology, Massachusetts General Hospital and Harvard Medical School, Boston, MA 02115, USA; 8Division of Epidemiology, Vanderbilt University Medical Center, Nashville, TN 37203, USA; william.j.blot@vumc.org; 9Division of Gastroenterology, Department of Medicine, Vanderbilt University Medical Center, Nashville, TN 37232, USA; 10Behavioral and Epidemiology Research Group, American Cancer Society, Atlanta, GA 30303, USA; lauren.teras@cancer.org; 11Department of Epidemiology, Johns Hopkins School of Public Health, and Department of Oncology, Johns Hopkins School of Medicine, Baltimore, MD 21205, USA; kvisvan1@jhu.edu; 12Department of Population Health, New York University Grossman School of Medicine, New York, NY 10016, USA; yu.chen@nyulangone.org (Y.C.); Anne.Jacquotte@nyulangone.org (A.Z.-J.); 13Brigham and Women’s Hospital, Boston, MA 02215, USA; 14Occupational and Environmental Epidemiology Branch, Division of Cancer Epidemiology and Genetics, National Cancer Institute, National Institutes of Health, Rockville, MD 20850, USA; berndts@mail.nih.gov; 15Centre for Public Health Research, Massey University, Wellington 6140, New Zealand; j.d.potter@massey.ac.nz; 16Department of Mathematics, Duke University, Durham, NC 27705, USA; 17Center for Genetic Epidemiology, Department of Preventive Medicine, University of Southern California and USC Norris Comprehensive Cancer Center, Los Angeles, CA 90089, USA; christopher.haiman@med.usc.edu; 18Department of Epidemiology & Population Health, Albert Einstein College of Medicine, Bronx, NY 10461, USA; sylvia.smoller@einstein.yu.edu; 19Division of Public Health Sciences, Fred Hutchinson Cancer Research Center, Seattle, WA 98109, USA; ltinker@whi.org; 20Infections and Cancer Epidemiology, German Cancer Research Center (DKFZ), 69120 Heidelberg, Germany; t.waterboer@dkfz-heidelberg.de (T.W.); j.butt@dkfz-heidelberg.de (J.B.)

**Keywords:** *Helicobacter pylori*, *Streptococcus gallolyticus*, colorectal cancer, sero-positivity, antibodies

## Abstract

Previously, we found that risk of colorectal cancer (CRC) is increased in individuals with serum antibody response to both *Helicobacter pylori* (HP) Vacuolating Cytotoxin (VacA) toxin or *Streptococcus gallolyticus* (SGG) pilus protein Gallo2178. In the present analysis, we tested the hypothesis that combined seropositivity to both antigens is a better indicator of CRC risk than seropositivity to single antigens. We used multiplex serologic assays to analyze pre-diagnostic serum for antibody responses from 4063 incident CRC cases and 4063 matched controls from 10 US cohorts. To examine whether combined SGG Gallo2178 and HP VacA sero-status was associated with CRC risk, we used conditional logistic regression models to estimate odds ratios (ORs) and 95% confidence intervals (CIs). Compared to dual sero-negative individuals, there was no increased risk for individuals sero-positive to SGG Gallo2178 only (OR: 0.93; 95% CI: 0.66–1.31) or to HP VacA only (OR: 1.08; 95% CI: 0.98–1.19). However, dual sero-positive individuals had a >50% increased odds of developing CRC (OR: 1.54; 95% CI: 1.16–2.04), suggesting an interaction between antibody responses to these two pathogens and CRC risk (p_interaction_ = 0.06). In conclusion, this study suggests that dual sero-positivity to HP VacA and SGG Gallo2178 is an indicator of increased risk of CRC.

## 1. Introduction

Previously identified potentially modifiable risk factors for colorectal cancer (CRC), the third most common cancer in the US [1], include diet, physical inactivity, obesity, and smoking. However, these lifestyle factors have generally been associated with only a moderately increased risk of CRC. The most well established risk factors for CRC, a first degree family history of CRC and a personal history of inflammatory bowel disease or colorectal adenoma, are not possible targets for primary prevention. 

A more recent area of research regarding CRC etiology is focused on the potential association between gastrointestinal bacteria and CRC risk [2,3]. Two of the bacterial species of interest in this context include *Streptococcus gallolyticus* subsp. *gallolyticus* (*S. gallolyticus*) (SGG) and *Helicobacter pylori* (*H. pylori*). Interest in the potential connection of *S. gallolyticus* with CRC began with the observation that patients with infective endocarditis caused by *S. gallolyticus* more often than expected presented with a concomitant intestinal adenoma or CRC [4]. Previously, we explored the association between SGG and CRC risk in our consortium of 10 prospective cohort studies in the US, comprising serum samples of 4063 CRC cases and 4063 matched controls. We found no overall association of antibody responses to pilus protein Gallo2178, a virulence factor important for adherence to epithelial tissues, with CRC risk in the cohort consortium (OR: 1.23; 95% CI: 0.99–1.52). However, a time analysis between blood draw and CRC diagnosis revealed a statistically significant 40% increased risk of developing CRC with antibody responses to *S. gallolyticus* Gallo2178 with a CRC diagnosis within 10 years after blood draw (95% CI: 1.09–1.79) [5]. In the European Prospective Investigation into Cancer (EPIC), a consortium of prospective cohort studies throughout Europe, a strong association between Gallo2178 sero-prevalence and CRC was also found (OR: 3.01; 95% CI: 1.49–6.08) in a population with a median follow-up time of just 3.4 years (range 0.4 to 8.5 years) [6].

*H. pylori* is well established as the main carcinogen in the development of non-cardia gastric cancer [7]. Several studies have investigated whether *H. pylori* might be involved in carcinogenesis in extra-gastric sites, including CRC. The results of individual studies have been inconsistent, but two meta-analyses reported an overall statistically significant positive association between *H. pylori* infection and CRC [8,9]. To thoroughly evaluate this newly found association between *H. pylori* protein-specific infection and CRC risk, we investigated the association in our large, prospective US cohort consortium, and found a statistically significant 11% increased risk of developing CRC with antibody responses to the *H. pylori* Vacuolating Cytotoxin A (VacA) [10]. Additionally, similar to SGG, we found *H. pylori* sero-positivity, and specifically VacA sero-positivity, to be much more common among African Americans and other non-whites; further, the association of VacA antibody levels with CRC was stronger among non-white populations. Thus, these findings suggest that the serological response to *H. pylori* proteins could serve as a marker for individuals at increased risk of developing CRC.

In the present analysis, we sought to investigate the combined association of host response to these two pathogens, *S. gallolyticus* and *H. pylori*, and the risk of CRC in a nested case-control design utilizing pre-diagnostic serum or plasma samples from 4063 incident CRC cases and 1:1 matched controls from ten prospective US cohorts. Our hypothesis was that host immune response to these two gastrointestinal pathogens combined would better characterize those at elevated CRC risk than an immune response to either alone.

## 2. Materials and Methods

### 2.1. Study Population

As previously described [5,10], we established a consortium of ten prospective cohorts: Campaign Against Cancer and Stroke (CLUE); Cancer Prevention Study-II (CPSII); Health Professionals Follow-up Study (HPFS); Multiethnic Cohort Study (MEC); Nurses’ Health Study (NHS); NYU Women’s Health Study (NYUWHS); Physicians’ Health Study (PHS); Prostate, Lung, Colorectal, and Ovarian Screening Study (PLCO); Southern Community Cohort Study (SCCS); and Women’s Health Initiative (WHI), to explore the association of host immune response to infection and colorectal cancer. This study was approved by the Institutional Review Boards of all collaborating institutions.

Each cohort contributed data and pre-diagnostic biospecimens collected at baseline from incident colorectal cancer cases (defined based on the International Classification of Diseases for Oncology (ICD-O-3), codes C180–189, C199, and C209) and an equal number of controls, matched on sex, self-reported race/ethnicity, date of birth, and date of blood collection. Using incidence density sampling, one control was chosen at random for each CRC case from the appropriate risk sets consisting of all cohort members who had provided blood specimens and who were alive and free of cancer (except non-melanoma skin cancer) at the time of diagnosis of the index case. Covariates to be considered as potential confounders collected from participating cohorts and harmonized included education, smoking, body mass index (BMI), CRC screening history, and family history of CRC.

Of the initial 8420 samples that were assayed by multiplex serology (4210 cases and 4210 controls), 294 (147 pairs) were excluded from this analysis due to technical issues of insufficient volume, pipetting errors, and/or invalid measurements due to insufficient bead counts (100 pairs), or because the pair was mismatched by self-reported race/ethnicity and/or sex (47 pairs). Thus, these analyses were performed with a final study population of 8126 (4063 cases and 4063 controls). Among this population, the median age at diagnosis was 73 years (range: 40–97 years), there were more women than men (63% vs 37%), cancer primarily occurred in the colon (84%), and the median follow-up time for cases was 7 years (range: <1–40.2 years).

### 2.2. Multiplex Serology

This study is based on previously collected multiplex serology data. As previously described, fluorescent-bead-based multiplex serology utilizing a Luminex flow cytometer (Luminex Inc., Austin, TX, USA) was applied to measure the amount of bound IgG/IgA/IgM serum antibody against 13 *H. pylori* and 9 *S. gallolyticus* proteins recombinantly expressed as glutathione-S-transferase (GST)-tagged fusion proteins [6,11,12,13,14]. The resulting quantitative measure is a median fluorescence intensity (MFI), measured on at least 100 beads per set per sample. Calculation of net MFI is performed by subtracting background values resulting from a bead-set loaded with GST-tag only in addition to a serum-free reaction.

To determine sero-positivity to individual *H. pylori* antigens, we applied antigen-specific MFI cutoffs as previously defined within this population and assured by visual inspection of percentile plots [10]. For overall *H. pylori* sero-positivity, we applied the previously established definition of being positive to 4 or more of the 13 *H. pylori* antigens assessed [13]. Individuals were designated as *H. pylori* VacA-positive (Hp VacA+) if they met the criteria of both *H. pylori*-positive and VacA-positive to ensure that antigen-specific seropositivity did not result from cross-reactive antibody responses from infection with other pathogens expressing homologous proteins [10].

For *S. gallolyticus*, we again applied antigen-specific cutoffs as previously determined for this population [5], which were defined arbitrarily through visual inspection of percentile plots at the approximate inflection point of the curve to dichotomize antibody responses as sero-positive and sero-negative, as described previously for other antigens [15,16,17]. In the present study, we assessed only *S. gallolyticus* Gallo2178 because it was previously shown that sero-positivity to this protein was the strongest *S. gallolyticus* marker associated with CRC risk [5,6].

### 2.3. Statistical Analysis

Study participant characteristics of age, sex, self-reported race/ethnicity, education, BMI, smoking status, family history of CRC, and personal history of ever colonoscopy/sigmoidoscopy were compared by combined Hp VacA and SGG Gallo2178 sero-positivity status, among controls only, using a chi-squared test for categorical variables and the t-test for the continuous variable of age.

To examine whether combined HP VacA and SGG Gallo2178 sero-status was associated with CRC risk, we created a new categorical variable with four mutually exclusive values: sero-negative to both (HP VacA-/SGG Gallo2178-); sero-positive to Hp VacA only (HP VacA+/SGG Gallo2178-); sero-positive to SGG Gallo2178 only (HP VacA-/SGG Gallo2178+); and sero-positive to both (HP VacA+/SGG Gallo2178+). We then used conditional logistic regression models to estimate odds ratios (ORs) and 95% confidence intervals (CIs) for CRC, with the dual sero-negative group (HP VacA-/SGG Gallo2178-) as reference. P-values below 0.05 were considered statistically significant. Study participant characteristics as described above were considered as potential confounders; only BMI and education were associated with the main exposure of interest, combined Hp VacA and SGG Gallo2178 sero-positivity, and thus were adjusted for in secondary models among those study participants who did not have missing values for these factors (n = 6794; 84% of the full study population). An interaction between sero-positivity to the two bacterial proteins and association with CRC risk was examined through the inclusion of a multiplicative interaction term assessed by the Wald Chi-Square test.

Because we believed, a priori, that individuals of different self-reported races/ethnicities could have a different immune response to both infections [18,19], we repeated the same analyses as above, but stratified by self-reported race/ethnicity.

Secondarily, as we had previously found the SGG Gallo2178 association with CRC to be strongest among individuals with less than 10 years of follow up between baseline blood draw and CRC diagnosis [5,11], we also performed stratified analyses by follow-up time (<10 years vs. ≥10 years between blood draw and diagnosis).

## 3. Results

Among controls, individuals sero-positive to both HP VacA and SGG Gall02178, compared to individuals sero-negative to both, were more likely to be older, African-American, and have a lower educational attainment and higher BMI (Table 1). To note, controls were matched to cases on age, self-reported race/ethnicity, and sex.

Most study participants were sero-negative to both antigens (65% of controls and 63% of cases), almost a third were sero-positive to HP VacA but sero-negative to SGG Gallo2178 (31% of controls and 32% of cases), and only a small fraction were sero-negative to HP VacA but sero-positive to SGG Gallo2178 (2% of controls and 2% of cases), or sero-positive to both (2% and 3%, respectively). We found a > 50% increase in the odds of developing CRC among individuals sero-positive to both, compared to sero-negative to both (OR: 1.54; 95% CI: 1.16–2.04) (Table 2). There was no increased risk for individuals sero-positive to SGG Gallo2178 only (OR: 0.93; 95% CI: 0.66–1.31) or to HP VacA only (OR: 1.08; 95% CI: 0.98–1.19). Inclusion of a multiplicative interaction term suggested an interaction between antibody responses to these two bacteria and CRC risk (p_interaction_ = 0.06). Stratification by self-reported race/ethnicity showed a particularly strong association between dual sero-positivity and CRC among African Americans (OR: 2.00; 95% CI: 1.15–3.48), a population with an overall higher proportion of individuals with antibody responses to both pathogens as well (10%). However, there was no suggestion of an interaction between antibody responses among African Americans alone (p_interaction_ = 0.61). Adjusting for education and BMI, among those participants with these variables available, did not change the findings (Table 2).

Secondary analyses did not suggest differences in these associations by time from blood draw to diagnosis (Appendix A).

## 4. Discussion

In this nested case-control study of over 4000 prospectively ascertained CRC cases and 1:1 matched controls, we found that dual sero-positivity to HP VacA and SGG Gallo2178, as compared to dual sero-negativity to both, was associated with a 54% increase in the odds of developing CRC. This association appeared to be stronger in African Americans, who showed a doubling of odds for CRC. Previously, in the same consortium, we had found overall associations between presence of antibodies to each of these specific antigens and risk of CRC, separately, although these individual associations were weaker (ORs of 1.11 and 1.23, respectively) [5,10] than the 1.54 found in the combined analysis, suggesting a synergistic effect. To note, in those prior analyses, the reference groups were different (Hp VacA- and SGG Gallo2178-, respectively), as were the exposures (Hp VacA+ and SGG Gallo2178+, respectively), all without consideration of sero-status to other antigens.

The underlying mechanisms for the potential etiological roles of *H. pylori* and *S. gallolyticus* in colorectal carcinogenesis remain elusive, especially regarding any potential interaction between the bacteria. As the colon is not the natural habitat of *H. pylori*, if the association is causal, the bacteria might contribute to colorectal carcinogenesis via indirect effects [20]. One possibility is that *H. pylori* infection in the stomach and associated alterations in the gastric environment re-shape the gut microbiome [21]. Two recent studies have reported co-occurrence of harmful gastrointestinal microbiota with *H. pylori* infection. Comparing 212 *H. pylori*-infected individuals with 212 controls from a population-based study in Germany, 16S rRNA sequencing revealed that *H. pylori* cases had higher fecal bacterial diversity, but also that those with the greatest stool antigen load of *H. pylori* were less likely to have four genera (*Bacteroides, Barnesiella, Fusicatenibacter,* and *Alistipes*) associated with health promotion [22]. Separately, in a prospective study in China of matched tissue and stool samples pre- and post-*H. pylori* eradication, *H. pylori* infection was significantly associated with microbial dysbiosis, and successful eradication increased abundance of probiotic bacteria [23].

It has also been hypothesized that bacterial commensals like *S. gallolyticus* might become pathogenic after the gut intestinal barrier integrity is disrupted, as is the case during colorectal tumorigenesis. Laboratory studies have shown that the altered molecular conditions in CRC tumor tissue enables *S. gallolyticus* to proliferate favorably and to have a pro-inflammatory and proliferative effect on CRC cells [24,25]. Specifically, the protein Gallo2178 (as well as Gallo2179) has been found to play a key role in adhesion, allowing *S. gallolyticus* to invade colon tissue after a lesion in the epithelium has formed [26]. Thus, another possibility for a mutual effect of *H. pylori* and *S. gallolyticus* on colorectal carcinogenesis is that infection with *H. pylori*, through yet unknown mechanisms, could lead to CRC precursors that then enable other bacteria like *S. gallolyticus* to infect the colon and exert pro-carcinogenic effects.

Although there is no previous literature specifically on the combined effect of host response to these two bacteria and CRC risk, there exists previous research that has repeatedly reported potential associations of CRC with bacteria residing in the gut [27]. The rationale for these associations include multiple hypotheses of mechanism, including inflammation, genotoxicity, and oxidative stress through, for example, bacterial production of pro-oxidative reactive oxygen and nitrogen species [2]. A limitation of the present study is the lack of data on timing, duration, and persistence of these infections, which prevents any conclusions about timing and sequence of events. Moreover, it is important to emphasize that this study is of antibodies, and thus host response to infection, and cannot differentiate current versus past infection status. Furthermore, all *H. pylori* strains harbor a vacA gene, but strains can vary considerably in levels of VacA production, VacA secretion, or VacA activity [28]. The serum anti-VacA antibody responses detected in this study probably occur most commonly in individuals infected with *H. pylori* strains producing high levels of type s1/m1 VacA (a form exhibiting the highest levels of toxin activity in cell culture assays and associated with increased gastric cancer risk), but the current serologic assay was not designed to discriminate between antibody responses to different forms of VacA.

Additionally, only a small fraction of individuals in this study were dual positive for Hp VacA and SGG Gallo2178, suggesting the clinical significance is not broadly relevant. Nonetheless, we believe the data presented here, along with prior research, suggests that further examinations of co-infections and CRC could deepen our understanding of the heterogeneity of CRC etiology, contribute to the development of biomarkers, and provide the opportunity for studies to consider the potential of bacteria-related interventions to reduce risk of this common cancer.

## Figures and Tables

**Table 1 microorganisms-08-01698-t001:** Baseline factors among controls, overall and by combined *H. pylori* VacA (HP VacA) and *Streptococcus gallolyticus* subsp. *gallolyticus* (*S. gallolyticus*) (SGG) Gallo2178 sero-status.

		Combined Hp VacA and SGG Gallo2178 Sero-Status			
	All Controls (n = 4063)	HP VacA- and SGG Gallo2178- (n = 2640)	HP VacA+ and SGG Gallo2178- (n = 1260)	HP VacA- and SGG Gallo2178+ (n = 73)	HP VacA+ and SGG Gallo2178+ (n = 90)
Sex, n (%)					
Female	2556	1709 (67)	738 (29)	53 (2)	56 (2)
Male	1507	931 (62)	522 (35)	20 (1)	34 (2)
Age, mean (SD) ^a^		63 (10)	64 (9)	62 (10)	67 (10)
Self-reported race/ethnicity, n (%) ^a^					
Whites	3067	2186 (71)	790 (26)	51 (2)	40 (1)
African American	399	160 (40)	210 (53)	6 (2)	23 (6)
Asian American	307	188 (61)	105 (34)	8 (3)	6 (2)
Latino	211	63 (30)	126 (60)	3 (1)	19 (9)
Unknown/multiracial/other	79	43 (54)	29 (37)	5 (6)	2 (3)
Education, n (%) ^a^					
<High School	468	214 (46)	228 (49)	5 (1)	21 (4)
High School/GED	823	506 (61)	287 (35)	11 (1)	19 (2)
>High School, other than college	183	131 (72)	47 (26)	2 (1)	3 (2)
Some College	845	563 (67)	251 (30)	14 (2)	17 (2)
College Graduate	756	533 (71)	189 (25)	18 (2)	16 (2)
Graduate School	946	667 (71)	246 (26)	20 (2)	13 (1)
BMI [kg/m^2^], n (%) ^a^					
<20	99	67 (68)	29 (29)	1 (1)	2 (2)
20- <25	1226	819 (67)	358 (29)	31 (3)	18 (1)
25- <30	1456	930 (64)	467 (32)	20 (1)	39 (3)
≥30	748	453 (61)	251 (34)	18 (2)	26 (3)
Smoking, n (%)					
Never	1853	1208 (65)	557 (30)	45 (2)	43 (2)
Former	1622	1079 (67)	494 (30)	19 (1)	30 (2)
Current	546	325 (60)	198 (36)	7 (1)	16 (3)
Family history of CRC, n (%)					
No	2638	1760 (67)	772 (29)	52 (2)	54 (2)
Yes	408	263 (64)	128 (31)	8 (2)	9 (2)
Ever had a colonoscopy or sigmoidoscopy, n (%)					
No	1554	983 (63)	504 (32)	32 (2)	35 (2)
Yes	1494	1001 (67)	430 (29)	29 (2)	34 (2)
Self-reported diabetes, n (%)					
No	3279	2133 (65)	1011 (31)	60 (2)	75 (2)
Yes	271	153 (56)	100 (37)	8 (3)	10 (4)

^a^*p* < 0.05 (Chi-Squared test for categorical variables, *t*-test for continuous variable age) for *H. pylori* VacA-positive (HP VacA+) and SGG Gallo2178+ category compared to *H. pylori* VacA-negative (HP VacA-) and SGG Gallo2178- category.

**Table 2 microorganisms-08-01698-t002:** SGG Gallo2178 and HP VacA combined status and colorectal cancer risk, overall and by self-reported race/ethnicity.

					Among Matched Pairs with BMI and Education Available					
	Controls n (%)	Cases n (%)	OR (95% CI) ^a^	P_interaction_	Controls n (%)	Cases n (%)	OR (95% CI) ^a^	P_interaction_	OR (95% CI) ^b^	P_interaction_
**Overall**										
HP VacA-/SGG Gallo2178-	2640 (65)	2563 (63)	1.00 (Ref)		2191 (65)	2128 (63)	1.00 (ref)		1.00 (ref)	
HP VacA+/SGG Gallo2178-	1260 (31)	1303 (32)	1.08 (0.98, 1.19)		1061 (31)	1091 (32)	1.07 (0.96, 1.20)		1.06 (0.95, 1.19)	
HP VacA-/SGG Gallo2178+	73 (2)	66 (2)	0.93 (0.66, 1.31)		67 (2)	58 (2)	0.89 (0.61, 1.28)		0.87 (0.60, 1.25)	
HP VacA+/SGG Gallo2178+	90 (2)	131 (3)	1.54 (1.16, 2.04)	0.057	78 (2)	120 (4)	1.63 (1.21, 2.20)	0.026	1.57 (1.16, 2.12)	0.029
**Whites**										
HP VacA-/SGG Gallo2178-	2186 (71)	2141 (70)	1.00 (Ref)		1764 (72)	1733 (70)	1.00 (ref)		1.00 (ref)	
HP VacA+/SGG Gallo2178-	790 (26)	820 (27)	1.06 (0.95, 1.19)		620 (25)	640 (26)	1.05 (0.92, 1.20)		1.04 (0.91, 1.19)	
HP VacA-/SGG Gallo2178+	51 (2)	50 (2)	1.00 (0.67, 1.50)		48 (2)	43 (2)	0.91 (0.59, 1.40)		0.89 (0.58, 1.37)	
HP VacA+/SGG Gallo2178+	40 (1)	56 (2)	1.45 (0.95, 2.19)	0.301	33 (1)	49 (2)	1.52 (0.97, 2.38)	0.146	1.48 (0.94, 2.33)	0.145
**African Americans**										
HP VacA-/SGG Gallo2178-	160 (40)	127 (32)	1.00 (Ref)		145 (40)	112 (31)	1.00 (ref)		1.00 (ref)	
HP VacA+/SGG Gallo2178-	210 (53)	229 (57)	1.38 (1.01, 1.88)		189 (53)	207 (58)	1.42 (1.03, 1.96)		1.43 (1.03, 1.99)	
HP VacA-/SGG Gallo2178+	6 (2)	5 (1)	1.02 (0.31, 3.41)		6 (2)	5 (1)	1.04 (0.31, 3.48)		0.94 (0.27, 3.25)	
HP VacA+/SGG Gallo2178+	23 (6)	38 (10)	2.00 (1.15, 3.48)	0.609	19 (5)	35 (10)	2.31 (1.26, 4.22)	0.520	2.45 (1.32, 4.54)	0.405
**Asian Americans**										
HP VacA-/SGG Gallo2178-	188 (61)	180 (59)	1.00 (Ref)		188 (61)	179 (59)	1.00 (ref)		1.00 (ref)	
HP VacA+/SGG Gallo2178-	105 (34)	114 (37)	1.14 (0.80, 1.62)		104 (34)	114 (37)	1.16 (0.82, 1.66)		1.19 (0.83, 1.70)	
HP VacA-/SGG Gallo2178+	8 (3)	4 (1)	0.54 (0.16, 1.83)		8 (3)	4 (1)	0.55 (0.16, 1.84)		0.54 (0.16, 1.90)	
HP VacA+/SGG Gallo2178+	6 (2)	9 (3)	1.72 (0.55, 5.35)	0.232	6 (2)	9 (3)	1.74 (0.56, 5.40)	0.240	1.61 (0.51, 5.05)	0.296
**Latinos**										
HP VacA-/SGG Gallo2178-	63 (30)	72 (34)	1.00 (Ref)		59 (29)	71 (34)	1.00 (ref)		1.00 (ref)	
HP VacA+/SGG Gallo2178-	126 (60)	112 (53)	0.78 (0.51, 1.19)		125 (61)	109 (53)	0.72 (0.47, 1.12)		0.72 (0.46, 1.13)	
HP VacA-/SGG Gallo2178+	3 (1)	5 (20)	1.39 (0.32, 6.01)		3 (1)	4 (2)	1.01 (0.22, 4.73)		1.05 (0.22, 5.04)	
HP VacA+/SGG Gallo2178+	19 (9)	22 (10)	1.02 (0.50, 2.09)	0.948	19 (9)	22 (11)	0.98 (0.48, 2.00)	0.738	1.00 (0.48, 2.06)	0.759

^a^ Conditional logistic regression model (controls matched to cases on sex, race, date of birth, and date of blood collection); ^b^ Conditional Logistic regression model with adjustment for education (≤HS or GED, >HS ≤college graduate, ≥college graduate), and BMI (<25, ≥25 < 30, ≥30 kg/m^2^).

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
