# Peer review of "Association of Combined Sero-Positivity to Helicobacter pylori and Streptococcus gallolyticus with Risk of Colorectal Cancer"

_microorganisms, 2020, doi:10.3390/microorganisms8111698_

Round 1

Reviewer 1 Report

In this work, authors examined whether sero-positivity to both Helicobacter pylori VacA toxin and Streptococcus gallolyticus pilus protein Gallo2178 is a better indicator of CRC risk than sero-positivity to single antigens. Their conclusion is that dual sero-positivity to HP-VacA and SGG Gallo2178 is an indicator of increased risk of CRC.

This study was very well designed, with high quality methods, a large number of cases, and a strong statistical analysis.

As the authors themselves declare in the discussion section, those findings have not a broadly relevant clinical significance (due to the low number of +/+ patients), but they are very interesting to highlight and better comprehend the potential of bacteria-related interventions to reduce risk of colon cancer.

Comments:

  • Please use only one formulation for HP VacA, also in tabs. i.e. always HP-VacA or always HP VacA

Abstract:

  • please use “:” instead of “,” after OR and 95%CI. i.e.: (OR: 0.93; 95% CI: 0.66-1.31)
  • Explicate all acronyms the first time they are used. i.e. line 50: Vacuolating cytotoxin A (VacA)

Main text

  • Lines 109-115: too  many brackets make the text difficult to read, please use ":" i.e. "As previously described [5, 10], we established a consortium of ten prospective cohorts: Campaign Against Cancer and Stroke (CLUE), Cancer Prevention Study-II (CPSII), Health Professionals Follow-up study (HPFS), Multiethnic Cohort Study (MEC), Nurses’ Health Study (NHS), NYU Women’s Health Study (NYUWHS), Physicians’ Health Study (PHS), Prostate, Lung, Colorectal, and Ovarian Screening Study (PLCO), Southern Community Cohort Study (SCCS) and Women’s Health Initiative (WHI) to explore the association of host immune response to infection and colorectal cancer”
  • Line 131: (range: 40–97 years)
  • Line 133 (range: <1–40.2 years)
  • Lines 191-197: please use “:” instead of "," after OR and 95%CI
  • Line 243: “ bacterial production of pro-oxidative reactive oxygen and nitrogen species [2]”.

Tab S1 line 4: HP-VacA+/SGG-Gallo2178+

Author Response

Comments:

  • Please use only one formulation for HP VacA, also in tabs. i.e. always HP-VacA or always HP VacA

Thank you for this suggestion.  We have replaced Hp-VacA and SGG-Gallo2178 with Hp VacA and SGG Gallo2178 throughout the manuscript and tables.

Abstract:

  • please use “:” instead of “,” after OR and 95%CI. i.e.: (OR: 0.93; 95% CI: 0.66-1.31)

We have now done this throughout the paper.

  • Explicate all acronyms the first time they are used. i.e. line 50: Vacuolating cytotoxin A (VacA)

We have now spelled out VacA the first time it is used, in the abstract.

Main text

  • Lines 109-115: too  many brackets make the text difficult to read, please use ":" i.e. "As previously described [5, 10], we established a consortium of ten prospective cohorts: Campaign Against Cancer and Stroke (CLUE), Cancer Prevention Study-II (CPSII), Health Professionals Follow-up study (HPFS), Multiethnic Cohort Study (MEC), Nurses’ Health Study (NHS), NYU Women’s Health Study (NYUWHS), Physicians’ Health Study (PHS), Prostate, Lung, Colorectal, and Ovarian Screening Study (PLCO), Southern Community Cohort Study (SCCS) and Women’s Health Initiative (WHI) to explore the association of host immune response to infection and colorectal cancer”

We have made this formatting change as suggested.

  • Line 131: (range: 40–97 years)

We have made this formatting change as suggested.

  • Line 133 (range: <1–40.2 years)

We have made this formatting change as suggested.

  • Lines 191-197: please use “:” instead of "," after OR and 95%CI

We have made this formatting change as suggested.

  • Line 243: “ bacterial production of pro-oxidative reactive oxygen and nitrogen species [2]”.

We have made this formatting change as suggested.

Tab S1 line 4: HP-VacA+/SGG-Gallo2178+

We have made this formatting change as suggested.

Reviewer 2 Report

The manuscript by Epplein and collaborators reports a serological study aimed to test the hypothesis that combined seropositivity to two antigens, VacA from Helicobacter pylori and Gallo2178 from Streptococcus gallolyticus, would represent a better indicator of colorectal cancer risk than seropositivity to single antigens. Serological assays were used to analyze antibody responses from 4,063 colorectal cancer cases and 1:1 matched control from 10 US cohorts. Results from this analysis support the initial hypothesis that dual sero-positivity to H. pylori VacA and S. gallolyticus Gallo2178 is an indicator of increased risk of colorectal cancer.

The study is straightforward and focused to the aim. It is well conducted, and results and interpretations are convincing. However, the authors report that in the study they took into consideration participant characteristics such of age, sex, race/ethnicity, education, and others. In my opinion the word “race” is improper and should be deleted throughout the manuscript and from supplementary material, “ethnicity” is more appropriate.

Author Response

Comment: In my opinion the word “race” is improper and should be deleted throughout the manuscript and from supplementary material, “ethnicity” is more appropriate.

We feel that "ethnicity" is too restrictive, and so have replaced all mention of "race" with the specifically accurate "self-reported race/ethnicity".